# The Greenland Firn Compaction Verification and Reconnaissance (FirnCover) Dataset, 2013-2019

Michael J. MacFerrin[1], C. Max Stevens[2,3,4], Baptiste Vandecrux[5], Edwin D. Waddington[2], Waleed Abdalati[1]

[1]Cooperative Institute for Research in Environmental Sciences (CIRES), University of Colorado, Boulder, CO, USA
[2]Department of Earth and Space Sciences, University of Washington, Seattle, WA, USA
[3]NASA Goddard Space Flight Center, Greenbelt, MD, USA
[4]Earth System Science Interdisciplinary Center, University of Maryland, College Park, MD, USA
[5]Geological Survey of Denmark and Greenland, Copenhagen, Denmark

*Correspondence to* Michael J. MacFerrin (michael.macferrin@colorado.edu)

**Abstract.** Assessing changes in the density of snow and firn is vital to convert volume changes into mass changes on glaciers and ice sheets. Firn models simulate this process but typically rely upon steady-state assumptions and geographically and temporally limited sets of field measurements for validation. Given rapid changes recently observed in Greenland's surface mass balance, a contemporary dataset measuring firn compaction in a range of climate zones across the Greenland ice sheet's accumulation zone is needed. To fill this need, the Firn Compaction Verification and Reconnaissance (FirnCover) dataset comprises daily measurements from 48 strainmeters installed in boreholes at eight sites on the Greenland ice sheet between 2013 and 2019. The dataset also includes daily records of two-meter air temperature, snow height, and firn temperature from each station. The majority of the FirnCover stations were installed in close proximity to automated weather stations that measure a wider suite of meteorological measurements, allowing the user access to auxiliary datasets for model validation studies using FirnCover data. The dataset can be found here: https://www.doi.org/10.18739/A25X25D7M (MacFerrin et al., 2021).

## Copyright Statement

## 1. Introduction

Mass loss from the Greenland ice sheet (GrIS) is currently one of the largest direct contributors to sea-level rise (IPCC, 2013), and the majority of that loss since the early 2000s has been due to significant increases in surface melt and runoff (Velicogna et al., 2014, van den Broeke et al., 2016; Mottram et al., 2019). In Greenland's accumulation zone, which covers approximately 80% of the ice sheet (Box et al., 2006), annual snow accumulation is buried and densifies until it becomes glacial ice (Bader, 1954; Benson, 1962; Herron and Langway, 1980). Greenland's firn layer can be up to ~70 m thick (Schwander et al., 1997).

The GrIS's firn layer has been the subject of recent research for multiple reasons. First, assessments of Greenland's total mass balance using altimetry products use satellite-derived measurements of surface height to assess ice sheet volume, but need to resolve the evolution of the firn's porosity before converting volume change into mass change (i.e. Zwally et al., 2011; Shepherd et al., 2012; Csatho et al., 2014; McMillan et al., 2016, Smith et al., 2020). Second, the firn is able to retain part of the meltwater generated at the surface and buffer sea level rise (Pfeffer et al., 1991). The firn's retention capacity depends on: i) the pore volume available for meltwater storage (Harper et al., 2012), which is decreasing (Vandecrux et al, 2019); ii) the firn's cold content, which is the energy required to bring the firn to the melting temperature (Vandecrux et al., 2020a); and iii) on the capacity for the meltwater to reach depths where retention is possible, which is for example reduced in presence of low-permeability near-surface ice slabs (Machguth et al., 2016, MacFerrin et al. 2019). Third, the firn impacts climate records preserved in ice cores. Bubbles of atmospheric gasses become trapped in closed pores at the firn-ice transition, and knowledge of the age of the firn at this bubble close-off depth is essential to accurately establish the chronology of past climate changes (Schwander and Stauffer, 1984; Schwander et al., 1997). In all these cases, knowledge of the firn's compaction rate is crucial, yet to date there are relatively few in situ measurements of firn compaction, and there is no single, widely accepted model to simulate it. In this paper, we present the Firn Compaction Verification and Reconnaissance (FirnCover) dataset, which comprises measurements of firn compaction, depth-density profiles, and temperatures from eight sites on the GrIS.

## 2.    Background

*Firn densification* characterizes a general increase of the firn's bulk density and encompasses multiple processes. *Firn compaction* refers specifically to the compression of the firn due to overburden stress. Firn compaction occurs due to processes operating at the grain scale such as grain boundary sliding, sintering mechanisms including dislocation creep and lattice diffusion, and plastic deformation (Herron and Langway, 1980; Morris and Wingham, 2014). *Meltwater refreezing* increases the firn density when surface meltwater or rain refreezes in the firn's pore space (e.g. Braithwaite et al., 1994; Reeh, 2008). This occurs primarily in the warmest regions of the ice sheet's accumulation area. The two above-mentioned phenomena are interconnected because meltwater refreezing releases latent heat and increases the firn temperature, which accelerates compaction of surrounding firn (Pfeffer and Humphrey, 1996; Humphrey et al., 2012). In the highest-elevation zones of the ice sheet, where firn densification mainly occurs through compaction, the compaction rate in the near-surface firn varies seasonally due to the fluctuating temperature; the deeper firn does not experience this seasonal variation in compaction rate (e.g. Herron and Langway, 1980; Arthern et al., 2010; Ligtenberg et al., 2011; Morris and Wingham, 2014). Long-term changes in climate, such as air temperature and accumulation rate, may take many decades before they affect compaction rates over the full depth of the firn column (Li and Zwally, 2015). In the percolation zone, the seasonal cycle in near-surface firn compaction rate is also present. However, the infusion of meltwater can change the compaction rate on much shorter timescales (days to weeks) as latent heat rapidly warms the firn, and rapid densification can occur when the refrozen meltwater fills the pore space. This firn may then compact more slowly in the future because of its higher density. In this realm, a single anomalous melt season can significantly affect the depth-density profile (Brown et al., 2012).

Numerous models have been developed to simulate firn compaction and densification on various time scales (e.g. Herron and Langway, 1980; Zwally et al., 2011, Arthern et al., 2010; Ligtenberg et al., 2011; Morris and Wingham., 2014). On yearly and longer time scales, firn depth-density profiles and compaction rates can be estimated reasonably well using the mean annual air temperatures and accumulation rates (Herron and Langway, 1980). These firn-model results can be used e.g. to simulate the long-term evolution of the firn-ice transition depth for ice-core studies (Goujon et al., 2003; Rasmussen et al., 2013). On shorter (monthly, daily, or sub-daily) time scales, firn models can be forced with weather data and/or outputs from regional climate models (RCMs) to simulate the firn temperature, density, and thickness change (e.g. Vandecrux, et al., 2020a). Results from these model runs can be used to correct repeat surface-elevation measurements from altimetry for firn changes (e.g. Smith et al., 2020). Numerous recent studies have coupled meltwater-percolation schemes to firn-compaction models (e.g. Reeh, 2008; Kuipers-Munneke et al., 2015; Vionnet et al., 2012; van Pelt et al., 2012; Verjans et al., 2019; Vandecrux et al., 2020a) to simulate liquid water content, refreezing, and runoff in the firn.

Most firn densification schemes have generally been developed using density profiles observed in firn cores (Herron and Langway, 1980; Sørensen et al., 2011; Kuipers-Munneke et al., 2015). By assuming that the firn is in steady state, a dated depth-density profile can be converted to a densification rate. There are several potential issues with this method. First, it is not necessarily safe to assume that the firn at a given site is in steady state. Even if the firn is in steady state, a compaction rate derived from the depth-density profile does not provide information about the firn's response to a transient climate or how its compaction rate varies on sub-annual timescales. Additionally, density profiles from the percolation zone cannot disentangle contributions of firn compaction and meltwater refreezing, which makes it difficult to assess these two processes in firn models. Finally, some densification models are tuned to match firn-density observations while forced by RCM-simulated surface forcing. The biases that may exist in the surface forcing are then compensated by the tuning of the densification model, which can then give inappropriate response under a different climate forcing.

Among the numerous firn models, none is broadly accepted as a definitive model. Lundin et al. (2017) showed that these models agree neither in steady-state nor in transient modes. Further, certain firn models are tuned specifically for Greenland or Antarctica, despite the fact that the physical processes driving densification should not vary solely due to geographic location. Vandecrux et al. (2020b) compared numerous firn-meltwater models to observations and found that while different models accurately simulated physical characteristics of different firn zones in Greenland, no single model accurately represented firn density, temperature and water content at all sites.

The uncertainties associated with firn-model development and the disagreement among the existing models underscore the need for direct measurements of firn compaction. The direct observation of firn compaction implies either tracking the thickness of a portion of firn (Hamilton et al., 1998; Arthern et al., 2010), the optical tracking of layers in a borehole (Hubbard

et al., 2020) or the tracking of layers in repeated high-resolution density profiles (Morris and Wingham, 2014). Most of the firn compaction measurements have been conducted in Antarctica (Hamilton et al., 1998; 2002; Arthern et al., 2010; Hubbard et al. 2020). Lastly, the only firn compaction measurements available in Greenland (Morris and Wingham, 2014) derived average compaction rates over specific periods spanning from 2004 to 2011 and over a single transect in central western Greenland.

To fill this knowledge gap and increase our understanding of firn densification in Greenland, we present data from the Firn Compaction Verification and Reconnaissance (FirnCover) project, which monitored firn compaction between 2013 and 2019 at eight stations on the GrIS. Each station monitors firn compaction with strainmeters installed over boreholes at various depth ranges, as well as firn temperature, air temperature and surface height. Additionally, we measured depth-density and stratigraphy profiles of recovered cores and in snow pits during each field visit. In this paper, we describe the FirnCover stations (Section 3) and the dataset organization (Section 4), and then we present a preliminary analysis of the dataset (Section 5).

## 3.    The FirnCover stations and dataset

The eight FirnCover stations are located in various climate zones of the ice sheet accumulation area (Figure 1, Table 1). Two stations, Summit and EastGRIP, are located in the high-elevation, dry-snow zone of the ice sheet, where melt rarely occurs and where firn compaction is the dominant densification process. Six stations are located in the percolation zone of the ice sheet, where changes in surface meltwater and refreezing are changing the structure and density profiles of firn (Machguth et al., 2016; Vandecrux et al., 2018; MacFerrin et al., 2019). The KAN_U, Dye-2, and EKT stations were installed in Spring 2013 and the remainder of the stations in Spring 2015. At every station, additional instruments were installed in new boreholes upon subsequent visits. The instruments were generally within 10 m of the tower and their position relative to the tower are given in the table Compaction_Instrument_Metadata (Table A7).

Each station included a suite of instruments, which we detail below, and was equipped with a tower to hold instrumentation, a data logger (Campbell CR800), a solar panel, and a battery. Borehole strain rates were recorded daily, while air temperature, surface height, and firn temperature measurements were recorded hourly. During most years, summary data from the instruments was transmitted from each station once per day using an Iridium short-burst data modem. Full data tables were saved on the data logger and were read from each station upon visits in the field, which usually occurred in late April or early May.

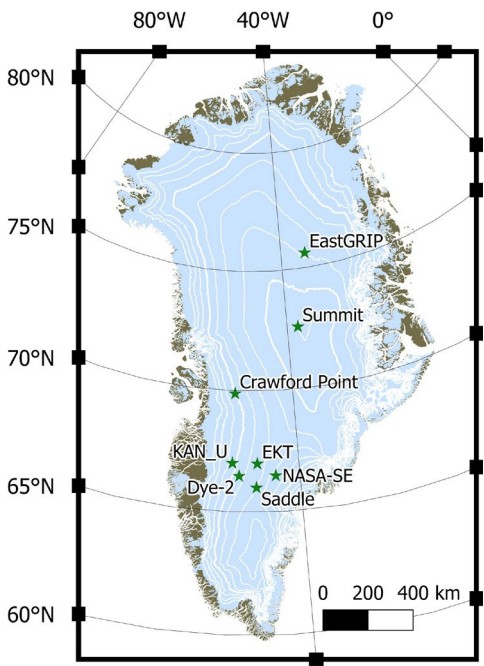

**Figure 1: FirnCover station locations in Greenland. White lines are 1000 m (thick) and 250 m (thin) elevation contours.**

**Table 1: FirnCover station locations and 2015-2017 average winter accumulation (Heilig et al., 2020).**

| Station name | Latitude (°) | Longitude (°) | Elevation (m) | Winter accumulation (mm w.e.) |
|---|---|---|---|---|
| KAN_U | 67.00 | -47.02 | 1840 | 249 |
| Dye-2 | 66.47 | -46.28 | 2119 | 329 |
| EKT | 66.99 | -44.39 | 2361 | 309 |
| Saddle | 66.00 | -44.50 | 2456 | 380 |
| NASA-SE | 66.48 | -42.50 | 2370 | 616 |
| Crawford Point | 69.88 | -46.99 | 1942 | 386 |
| Summit | 72.58 | -38.50 | 3208 | 218 |
| EastGRIP | 75.63 | -35.94 | 2666 | 324 |

### 3.1.    The FirnCover strainmeters

The main components of each FirnCover station were borehole strainmeters, which made daily measurements of borehole

lengths. These used the "coffee-can" method (Hulbe and Whillans, 1994; Hamilton et al., 1998) to continuously monitor firn compaction, similar to the method used by Arthern et al. (2010). Each instrument was composed of a line with a weight attached to one end and connected to a spring-loaded potentiometer on the other end. The weight was anchored at the bottom of a borehole, and the potentiometer was placed at the top of the borehole. As the borehole shortened due to firn compaction,

the potentiometer reeled in the string to maintain tension (Figure 2), and a data logger recorded the length of string that had
been reeled in. We here present data from 48 strainmeters installed at 8 FirnCover stations. Table 2 lists metadata for each
instrument, including the initial depths of the boreholes.

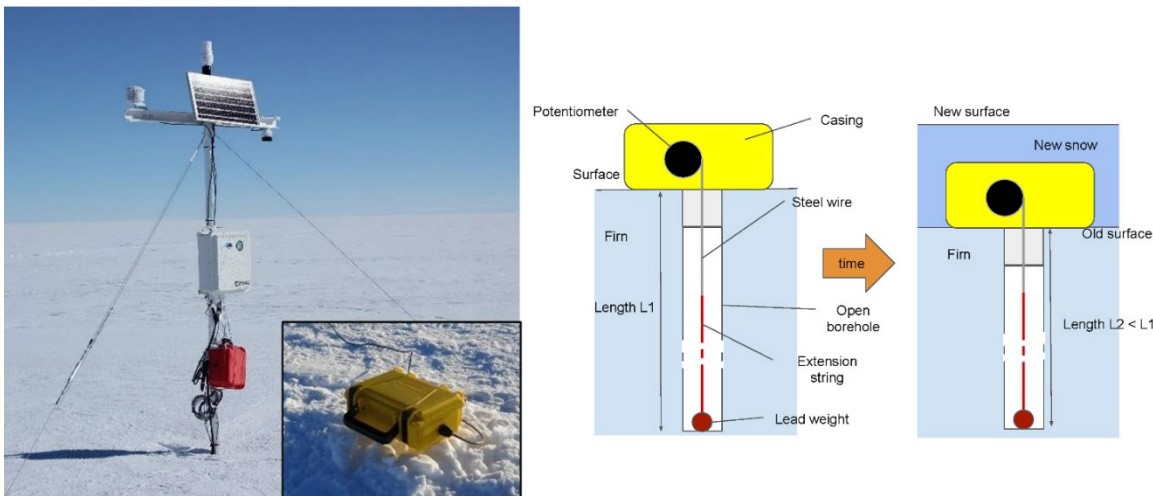

**Figure 2: FirnCover station (left), strainmeter casing (inset), and strainmeter conceptual design (right)**

The potentiometers were high-precision analog HX-PA units from Unimeasure, Inc. (Bend, Oregon). The end of the
potentiometer's steel wire was attached to a Vectran string that extends to the bottom of the borehole. The string was anchored
using a 0.226 kg lead weight. Each potentiometer was independently calibrated before installation. Including the potentiometer
accuracy and a minimal elongation of the extended string, measurement uncertainty on the borehole length is within ± 2 cm.
Measurement of the borehole shortening, however, is insensitive to the elongation of the wire that is under a constant load and
can be made with an accuracy of ±2 mm. The potentiometer was enclosed in a weatherproof plastic case with an opening at
the bottom. To stabilize the instrument atop the borehole, it was installed atop a 0.61 m$^2$ white PVC plastic platform. A section
of PVC pipe (0.1-0.7 m long) was attached to the bottom of the casing and inserted in the top of the borehole to prevent the
collapse of the top of the borehole and keep the instrument in place. The line lowered in the borehole was covered with
hydrophobic lithium grease to prevent water from refreezing on it and to keep the line from snagging on the instrument or
freezing to the side of the borehole.

To install each instrument, a borehole was drilled into the snow and firn using a Kovacs (diameter 9 cm) coring drill. The
weighted Vectran string was then lowered into the borehole, and the potentiometer platform was placed atop of the borehole
(Figure 2). The length of the string was set so that the potentiometer's steel cable was near its full extension, maximizing the
distance over which the borehole shortening could be observed. Some instruments were installed on the surface and thus

measured both the compaction of near-surface snow and underlying firn. Other instruments were installed at the bottom of snow pits, beneath the annual layer of snow, to measure the compaction of the underlying firn only. In dry-snow regions (Summit, EastGRIP) all instruments were installed directly on the surface, while instruments in the percolation zone were mixed between surface and snow-pit installations (see non-zero initial depth of borehole top in Table 2). The depth of each borehole was measured both along the core (by reassembling core segments on the surface) and by using the Vectran line to directly measure the borehole. Instruments #1-10, installed in 2013, use the approximate core length (as borehole length was not measured); the remaining instruments use the measured borehole length. Borehole and core-length measurements typically agreed to within 0-8 cm.

**Table 2: FirnCover instrument metadata.**

| Site | Instrument ID | Recording start date | Recording end date | Initial Depth of borehole top (m) | Initial Depth of borehole bottom (m) |
|---|---|---|---|---|---|
| Crawford Point | 22 | 27 May 2015 | 10 October 2018 | 1.03 | 17.33 |
| | 23 | 26 May 2015 | 10 October 2018 | 0 | 2.1 |
| | 24 | 27 May 2015 | 10 October 2018 | 1.09 | 9.38 |
| | 25 | 27 May 2015 | 10 October 2018 | 1.13 | 5.17 |
| | 42 | 17 May 2016 | 10 October 2018 | 0 | 18.09 |
| | 48 | 23 May 2017 | 10 October 2018 | 0 | 22.3 |
| Dye-2 | 4 | 09 May 2013 | 4 September 2019 | 0 | 2 |
| | 5 | 09 May 2013 | 4 September 2019 | 1.35 | 11.35 |
| | 6 | 09 May 2013 | 4 September 2019 | 1.35 | 17.35 |
| | 21 | 21 May 2015 | 4 September 2019 | 0.85 | 18.85 |
| | 39 | 10 May 2016 | 4 September 2019 | 0 | 17.3 |
| | 47 | 11 May 2017 | 4 September 2019 | 0 | 22.85 |
| EKT | 7 | 19 May 2013 | 4 September 2019 | 0 | 2 |
| | 8 | 19 May 2013 | 4 September 2019 | 1.25 | 6.25 |
| | 9 | 19 May 2013 | 4 September 2019 | 1.25 | 11.25 |
| | 10 | 19 May 2013 | 4 September 2019 | 1.25 | 17.25 |
| | 12 | 8 May 2015 | 4 September 2019 | 0.9 | 14.9 |
| | 36 | 3 May 2016 | 31 July 2019 | 0 | 17.95 |
| | 44 | 5 May 2017 | 4 September 2019 | 0 | 22.24 |
| EastGRIP | 26 | 28 May 2015 | 10 October 2018 | 0 | 15.83 |
| | 27 | 28 May 2015 | 10 October 2018 | 0 | 4.12 |

| | | | | | |
|---|---|---|---|---|---|
| | 28 | 29 May 2015 | 10 October 2018 | 0 | 8.05 |
| | 29 | 29 May 2015 | 10 October 2018 | 0 | 15.53 |
| | 40 | 16 May 2016 | 10 October 2018 | 0 | 16.28 |
| | 49 | 18 May 2017 | 10 October 2018 | 0 | 20.38 |
| KAN_U | 1 | 30 April 2013 | 9 May 2019 | 1.2 | 6.2 |
| | 2 | 30 April 2013 | 9 May 2019 | 0 | 2 |
| | 3 | 30 April 2013 | 9 May 2019 | 1.2 | 20.5 |
| | 11 | 05 May 2015 | 9 May 2019 | 0.64 | 14.14 |
| | 35 | 29 Apr 2016 | 9 May 2019 | 0 | 16.51 |
| | 43 | 28 Apr 2017 | 9 May 2019 | 0.78 | 22.86 |
| NASA SE | 13 | 12 May 2015 | 28 May 2018 | 0 | 16.4 |
| | 14 | 12 May 2015 | 28 May 2018 | 0 | 2.05 |
| | 15 | 12 May 2015 | 20 May 2018 | 0 | 8 |
| | 16 | 12 May 2015 | 28 May 2018 | 0 | 16.2 |
| | 45 | 6 May 2017 | 28 May 2018 | 0 | 22.17 |
| Saddle | 17 | 16 May 2015 | 31 Aug 2017 | 0 | 16.1 |
| | 18 | 16 May 2015 | 31 Aug 2017 | 0 | 2.03 |
| | 19 | 16 May 2015 | 31 Aug 2017 | 0 | 8.17 |
| | 20 | 16 May 2015 | 31 Aug 2017 | 0 | 16.3 |
| | 38 | 6 May 2016 | 31 Aug 2017 | 0 | 18.53 |
| | 46 | 8 May 2017 | 31 Aug 2017 | 0 | 22.34 |
| Summit | 30 | 29 May 2015 | 7 October 2018 | 0 | 15.73 |
| | 31 | 29 May 2015 | 7 October 2018 | 0 | 4.23 |
| | 32 | 29 May 2015 | 7 October 2018 | 0 | 7.77 |
| | 33 | 30 May 2015 | 7 October 2018 | 0 | 15.79 |
| | 41 | 17 May 2016 | 7 October 2018 | 0 | 16.08 |
| | 50 | 21 May 2017 | 7 October 2018 | 0 | 21.99 |

### 3.2. Air temperature, surface height and firn temperature observations

Each FirnCover station was equipped with a Campbell L109 air-temperature thermistor with 6-plate radiation shield, which measured air temperature hourly at approximately 2 m ground height. Snow-surface height was measured from 2015 onward with a SR50 sonic-ranging sensor mounted on the tower cross-beam. A string of 24 10-K$\Omega$ resistance-temperature diodes

(RTDs, from Omega sanitary, Class A, IEC 60751 standard) measured firn temperatures from 0 to approximately 14 m depth (every 0.5 m from 0-10 m depth, every 1 m thereafter). The manufacturer-stated precision of the RTDs is ±0.2°C. Some RTD-string boreholes were less than 14 m due to accumulated drill shavings at the bottom of the boreholes. RTD measurements are corrected for wire resistance (by measuring across a 25[th] bare wire without an RTD), and measured resistances are converted to temperature using formulae provided by the RTD manufacturer. The RTD strings were installed in separate boreholes that were backfilled with snow. The initial installation depths of each RTD string are noted in the FirnCover_Station_Metadata data table (Table A5). The daily depth of each thermistor is calculated by adding the original installation depth to the snow depth measured by the sonic ranging sensor. Unlike air temperature, surface height and firn temperature are available as daily averages.

### 3.3.    Firn core and snow pit observations

Firn cores were retrieved from each of the FirnCover strainmeter boreholes. To understand the structure of the firn at each FirnCover instrument, the cores were visually inspected for stratigraphic layers (ice lenses, etc.) at ~1 cm vertical resolution, and cut into segments to measure density at ~10 cm resolution (Figure 3). Density profiles from all cores logged by FirnCover field campaigns are included in NASA's SUMup dataset (Koenig and Montgomery, 2019; Montgomery et al., 2018).

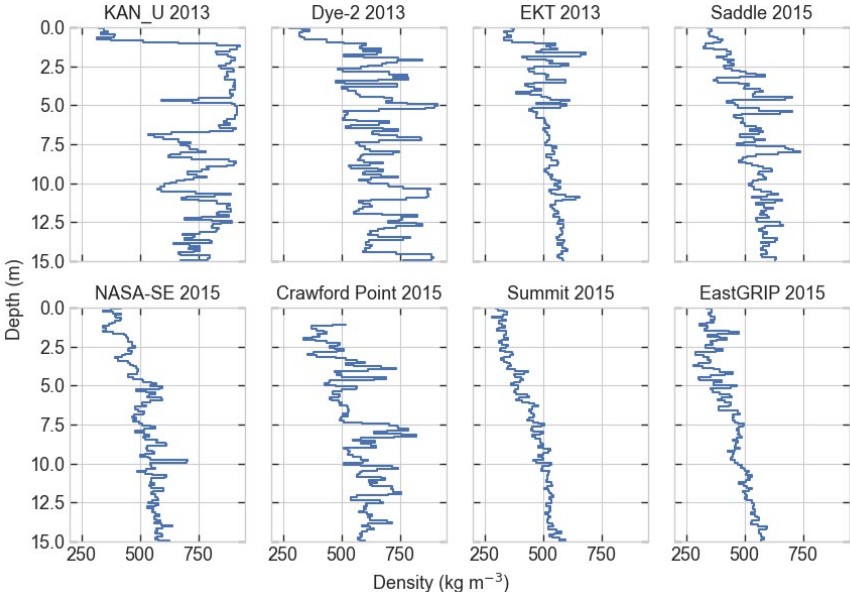

**Figure 3. Firn density profiles at the first visit of each site. These and other density profiles from FirnCover are available from Koenig and Montgomery (2019).**

**4. Dataset structure and handling**

The FirnCover dataset is organized in a single .hdf5 file, which comprises four data tables and three metadata tables. Table 3 gives a summary of the data tables, and tables A1 to A7 detail the variables contained in each table.

**Table 3: Overview of the FirnCover data tables**

| Table name | Content | Further details in |
|---|---|---|
| Compaction_Daily | site name, daily timestamp, instrument ID, compaction ratio, potentiometer wire correction ratio, potentiometer cable length, compaction borehole length, top and bottom depth | Table A1 |
| Air_Temp_Hourly | Site, hourly timestamp, air temperature | Table A2 |
| Meteorological_Daily | site name, daily timestamp, battery minimum and maximum voltage, panel mean temperature, air hourly minimum, median and maximum temperature, sonic ranger quality raw and corrected distance, raw and interpolated snow depth | Table A3 |
| Firn_Temp_Daily | site name, daily timestamp, thermistor average and maximum resistance value, uncorrected and corrected temperature value, average resistance of the cable used for correction, depth of the sensors | Table A4 |
| Station_Metadata | site name, iridium URL, latitude, longitude, installation date, comments, thermistor string number, thermistor installation date, number of thermistors usable, depths at installation, direction and distance from tower | Table A5 |
| Station_Visit_Notes | site name, date of visit, notes from each visit. | Table A6 |
| Compaction_Instrument_Metadata. | instrument ID, site name, installation date, borehole top and bottom depth from surface, initial length, direction | Table A7 |

| | and distance from tower, borehole ID in SUMup firn density dataset | |
|---|---|---|

At most strainmeters, the first weeks to months of record show relatively high compaction rates. This initial period of increased compaction is more pronounced for instruments installed at the surface than the ones installed at the bottom of a snow pit (Table 2). We consider these high initial compaction rates to be the result of the instrument settling over the snow and firn. This period of initial settling needs to be discarded to study the firn after it adapted to the presence of the instrument. At KAN_U, where the deeper firn is rich in ice (Figure 3), settling of the instrument is mainly due to the surface snow and took about a month. At Summit, where the firn has no ice layers, settling took about two months. For this preliminary analysis, we discard the first 60 days of recordings for each instrument, but a site-specific analysis of instrument settling may allow the recovery of more observations within that period.

Some compaction data was read directly from the station's data logger in 32-bit floating point format. For measurements where data tables were unable to be directly read due to lack of re-visit, data summaries from Iridium transmissions were used with 16-bit floating point values. Due to the limited data resolution, borehole lengths recorded from Iridium transmissions exhibit a 2 mm stepwise discretization rather than smooth continuous measurements. This can influence compaction rates when computed as derivatives of borehole lengths over time. In the present analysis, we use a two-month-wide running mean to smooth the borehole length. This filtering removes most of the noise, but it may also smooth part of seasonal changes of compaction rates. The dataset includes the unfiltered data, and we recommend that users apply their own filtering strategy specific to their needs.

Four of the stations had periods when the entire station was not recording data. These were: Summit, from 21-May-2017 to 23-August-2017; EKT, from 11-May-2017 to 18-May-2018; KAN_U from 07-November-2017 to 30-April-2018 and from 13-January-2019 to 20-February-2019; and Saddle from 30-May-2015 to 05-May-2016. For a number of the instruments, there are periods of data that we consider suspicious because of abrupt jumps in the compaction rates. We hypothesize that this could be due to ice buildup on the cable that prevented the instrument from working, and once the cable became free the instrument began to work again. The suspicious measurements are listed in Table 4. We exclude these suspicious data from our analysis in Section 5. For transparency, they are still included in the released dataset, but we advise caution when using them.

**Table 4. Periods with suspicious recordings removed from the analysis.**

| Instrument ID | Failure start date | Failure end date |
|---|---|---|
| 13 | 20 February 2018 | - |

| | | |
|---|---|---|
| 10 | 29 July 2019 | - |
| 42 | - | 14 November 2017 |
| 48 | - | 18 November 2017 |
| 48 | 27 May 2018 | 19 July 2018 |
| 1 | - | 1 December 2013 |
| 35 | - | 1 September 2016 |
| 43 | 16 July 2018 | - |

## 5.    Data overview and preliminary analysis

Figure 4 shows the change in borehole length measured by each potentiometer at the eight sites. NASA-SE shows the steepest borehole shortening with instruments #36 installed in 2017 and also the largest change in borehole length, -1.25 m for the 16.2 m long borehole #16 installed in 2015. This rapid shortening of the borehole largely stems from the climatic conditions at NASA-SE because: (1) high accumulation rates create a thick, low-density layer near the surface (Figure 3) which compact faster than high density snow, and (2) the fast build-up of new snow atop the borehole increases overburden pressure quickly,

which speeds the densification rate. At Dye-2 and KAN_U, the borehole shortening is the least pronounced. This is likely due to higher air temperatures, higher melt and lower snowfall at these sites; together they lead to higher firn density and ice content which decreases compaction rate (Figure 3). At the other sites, total borehole shortening ranges from a few centimeters to about a meter at EKT depending on the climatic conditions and the length of the observation period. Most sites, but especially Summit and EastGRIP, show a seasonality in the borehole shortening rate: boreholes shorten faster (steeper curve in Figure 4)

during and after summer months (orange shaded areas) and slower (flattening of curves in Figure 4) in the winter/spring months.

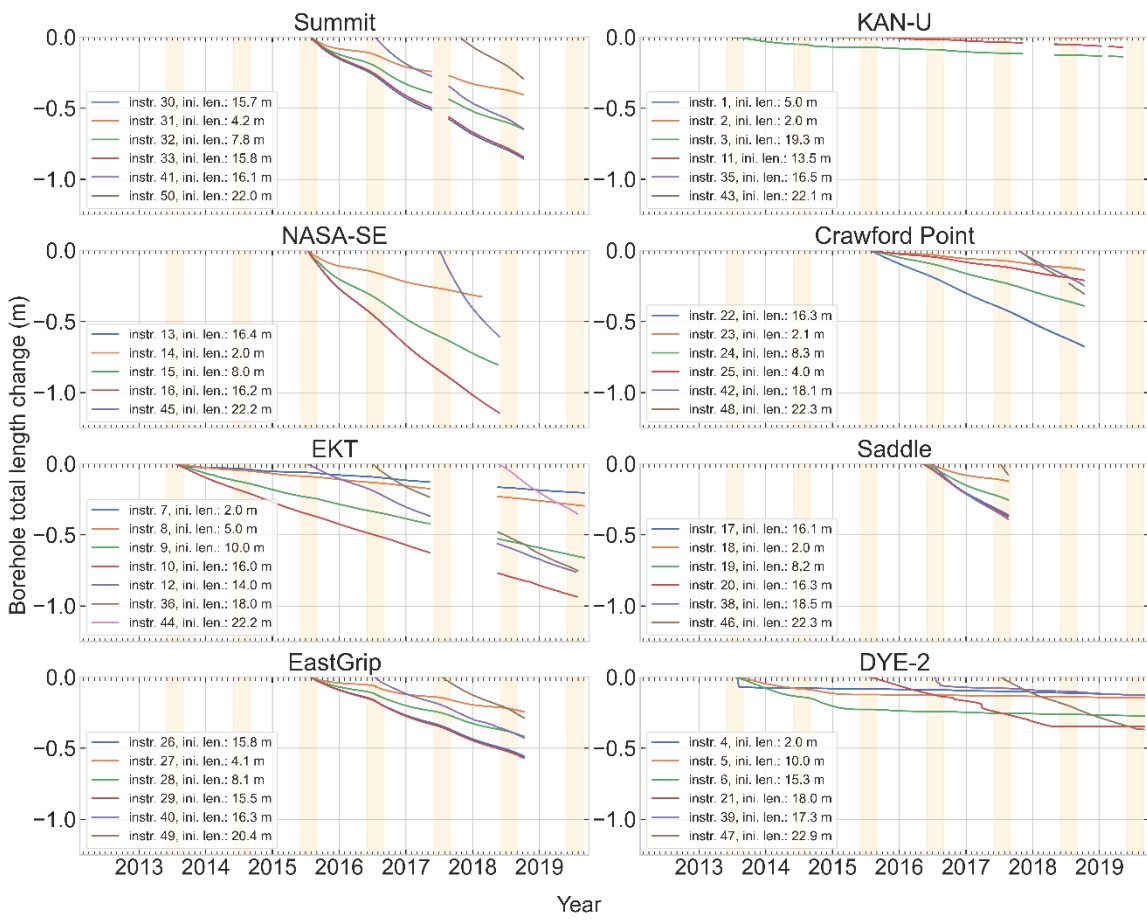

**Figure 4. Borehole length changes. The legend indicates the ID of each instrument as reported in Table 2 and the initial length of each borehole. June-July-August are highlighted in orange.**

It is possible that our compaction measurements could be affected by horizontal divergence (Horlings et al., 2021). However, for the present analyses, we consider these effects to be negligible, which is consistent with firn-densification modeling efforts in Greenland (Kuipers-Munneke et al., 2015). A more thorough analysis could use ice velocity measurements (e.g. Joughin et al., 2016) to explicitly account for the effects of ice flow.

The difference between sites can be further investigated by looking at the compaction rates, which are calculated by taking the time derivative of the borehole length data (Figure 5). As discussed above, NASA-SE shows the largest daily changes and KAN_U the lowest. At each site, instruments installed in deeper boreholes show larger magnitude of daily compaction compared to shorter instruments. The faster compaction after the installation of the instrument appears as large initial firn

compaction rates in Figure 5. Faster compaction during the first summer following the installation of the instruments also

stems from the conduction of warmer surface temperatures down to the instrument. These warmer firn temperatures during summer increase the firn compaction rates. As mentioned previously, daily compaction rates at KAN_U are lower than at other sites due to the presence of a ~5 m thick ice slab at that site. The seasonality of the daily compaction rates is clearly visible at the dry snow sites, Summit and EastGRIP, but also at sites in the percolation areas: NASA-SE, Crawford Point , EKT and

Saddle. Daily compaction rates peak in the autumn and reach a minimum at the end of the winter (Figure 5). The delay between the highest surface temperatures in summer and the highest compaction rate is due to the time the surface temperatures need to diffuse down to the depth of the firn that the instrument is measuring.

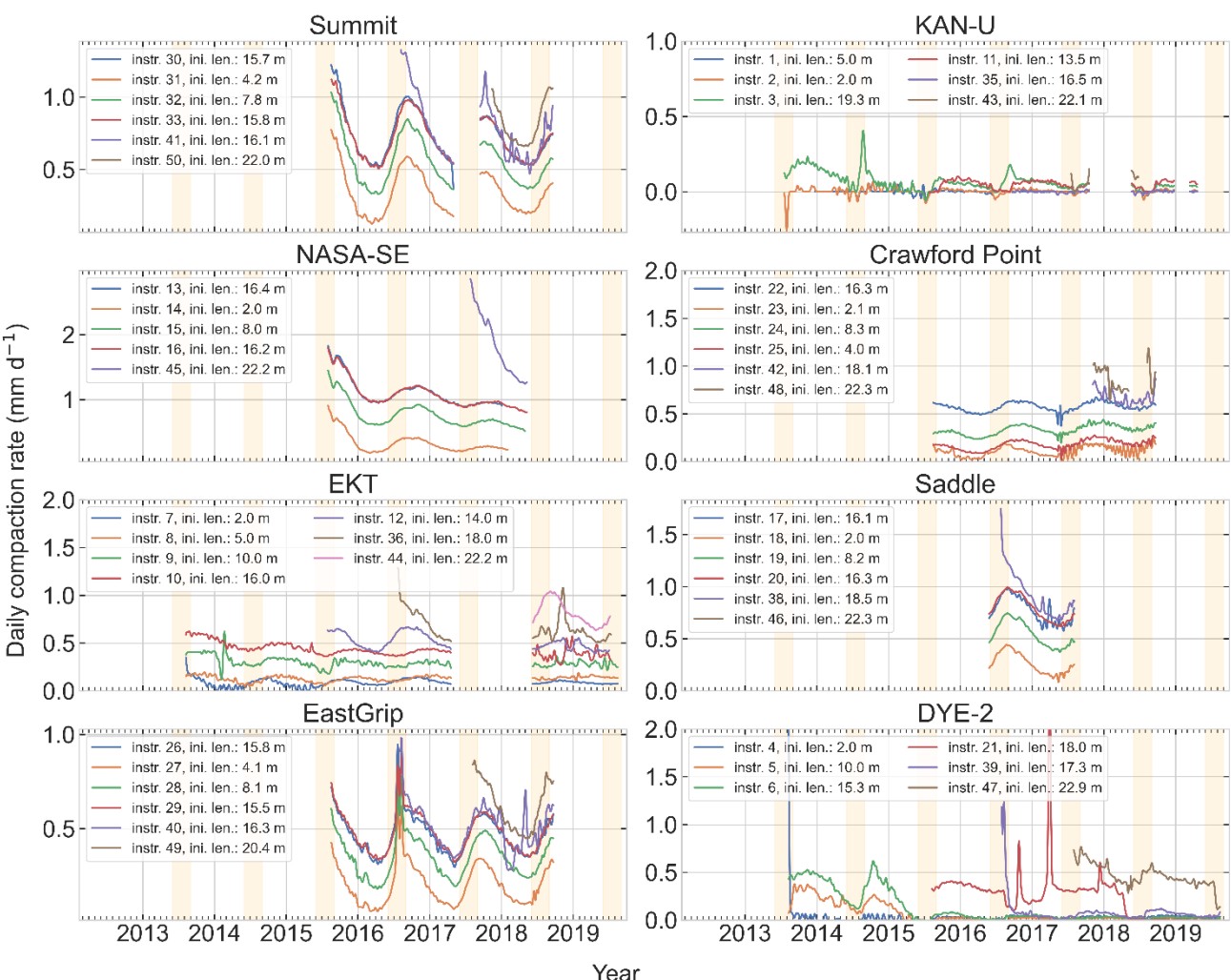

**Figure 5. Smoothed daily compaction rates. The legend indicates the ID of each instrument as reported in Table 2 and the initial length of each borehole. June-July-August are highlighted in orange. Note the different y-axes.**

The FirnCover dataset also includes measurements of air temperature, surface height, and, at all sites except EastGRIP and NASA-SE, firn temperature; these data enable us to relate the compaction rates (Figures 4 and 5) to the surface and subsurface conditions (Figures 6 and 7).

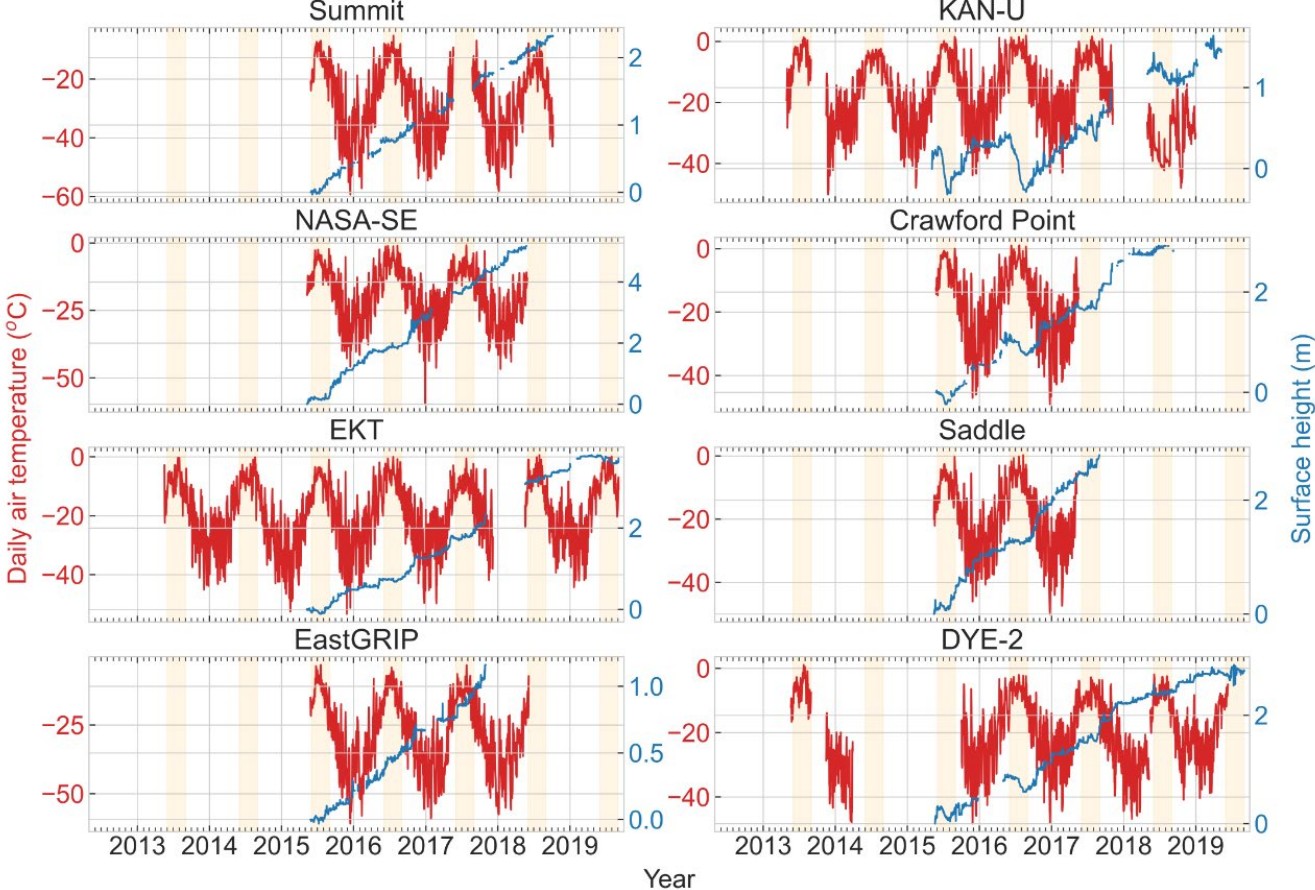

**Figure 6. Daily air temperature (red line, left axis) and surface height (blue line, right axis). June-July-August are highlighted in orange. Note the different y-axes.**

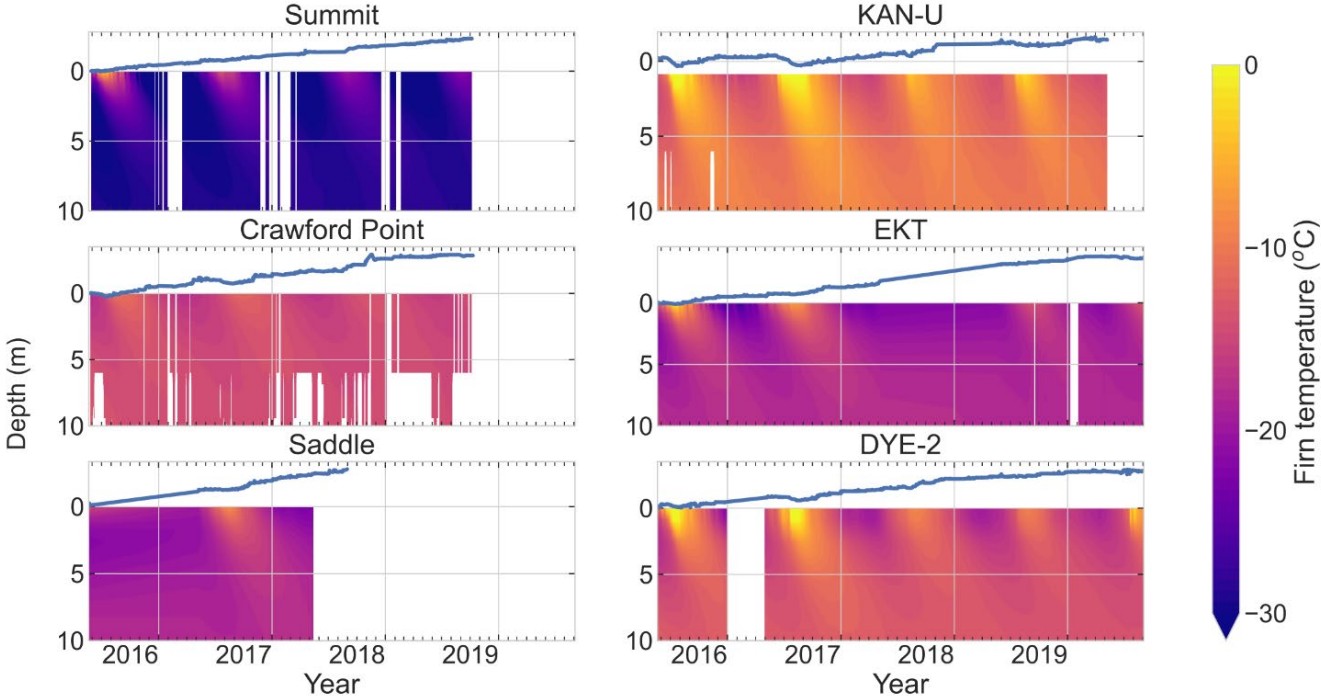

**Figure 7. Firn temperatures (interpolated) and surface height (blue solid line) observed at the FirnCover stations.**

The firn temperature measurements, in particular, allow analyses using the actual firn conditions rather than using average air temperature as a proxy for firn temperature, which is commonly done. For each site, we interpolate the 10 m firn temperature and average it over the 2015-2019 period covered by the measurements (Table 4). We compare this firn temperature to the average air temperature calculated for the years where more than 90% of the temperature measurements are available (Table 4). The average air temperature and interpolated 10m firn temperatures are rarely equivalent (Table 4). At Summit, the 10m firn temperatures are 2.6 °C lower than the average air temperature. This is due to strong near-surface atmospheric inversion and radiative cooling of the surface (e.g. Miller et al., 2017). At all the other sites, the 2 m firn temperature is higher than the average air temperature. We attribute this to meltwater percolation and latent heat release at depth (e.g. Pfeffer and Humphrey, 1996; Humphrey et al., 2012). This difference is largest at KAN_U where the firn is 7°C warmer than the average air temperature. At Saddle, the firn temperature is within a degree of the average air temperature. We interpret this as the neutralization of the two processes mentioned above: heat loss through radiative cooling at the surface and latent heat release during meltwater refreezing. This site-specific difference between 10 m firn temperature and average air temperature shows the limitation of firn compaction parameterizations that use air temperature as a proxy for firn temperature and how these parameterizations perform outside of their training site.

**Table 4. Average 10 m firn temperatures for the 2015-2019 period along with average air temperature at each site and difference between the two. Only years that have more than 90% available temperature readings are used for the average.**

| Site | 2015-2019 average 10 m firn temperature (ºC) | Average air temperature (ºC) | Years used for the average air temperature calculation | Difference (ºC) |
|---|---|---|---|---|
| Summit | -28.8 | -26.2 | 2016 | -2.6 |
| KAN_U | -9.5 | -16.6 | 2014 - 2016 | 7.1 |
| Crawford Point | -13.9 | -16.2 | 2016 | 2.3 |
| EKT | -17.9 | -20.2 | 2014 - 2017 | 2.3 |
| Saddle | -17.9 | -18.1 | 2016 | 0.2 |
| DYE-2 | -13.3 | -19.2 | 2016 - 2018 | 5.9 |

## 6.    Summary remarks

We present data from 48 strainmeters installed at eight sites located in different climatic zones of the Greenland ice sheet and covering the 2013-2019 period. Additional surface and firn measurements available at each of the FirnCover sites are firn density, air temperature, surface height and firn temperatures. These data will allow future work to investigate the interannual and seasonal response of firn compaction to surface and subsurface conditions. We also note that several other measurements are available at some of the FirnCover sites: at KAN_U the PROMICE automatic weather station has been operating since 2009 (Fausto et al., 2021); at Crawford Point, Saddle, NASA-SE, Summit and Dye-2, GC-Net weather stations document the history of these sites back to the 1990s and are still operating (Steffen et al., 1996). At Summit, extensive instrumentation is measuring the atmospheric conditions and the surface energy budget (e.g. Miller et al., 2017). At Dye-2, upward looking Ground Penetrating Radar (Heilig et al., 2018) and time-domain resistivity probes (Samimi et al., 2020) are also available for the 2016 melt season to detail meltwater percolation. These measurements, combined with the FirnCover compaction data, potentially allow investigations of how meltwater affects firn compaction. The FirnCover dataset will help to evaluate and calibrate firn models and help reduce uncertainty when using these models to interpret satellite altimetry measurements or calculating the past, current and future mass balance of polar ice sheets. The dataset can be found here: https://www.doi.org/10.18739/A25X25D7M.

## 7.    Data Availability

The FirnCover dataset is available at https://www.doi.org/10.18739/A25X25D7M (MacFerrin et al., 2021). The firn density profiles at the firn cover sites are available here: https://doi.org/10.18739/A26D5PB2S (Koenig and Montgomery, 2019).

## 8. Code Availability

All the scripts used to load, process and plot the FirnCover dataset are available here: https://doi.org/10.5281/zenodo.5854253 (Vandecrux et al., 2022).

## 9. Acknowledgements

The majority of this work, including instrumentation and station visits, was funded by NASA awards NNX15AC62G and NNX10AR76G. All authors acknowledge the work and efforts of multiple field logistic partners and team members for their essential help in maintaining the datasets and instruments: K. Alley, C. Charalampidis, W. Colgan, F. Covi, A. Crawford, M. Eijkelboom, S. Grigsby, A. Heilig, D. Hill, H. Machguth, S. Marshall, A. Rennermalm, S. Samimi, T. Snow, A. Sommers, D. van As, the Summit Station scientific team, the EastGRIP team and Polar Field Services. Lastly, we thank Ian McDowell, Joel Harper and Megan Thompson-Munson for constructive comments on the manuscript.

## 10. Author contribution

MM conducted the conceptualization, funding acquisition, the methodology, the field investigation and the data curation. CMS participated to the conceptualization, funding acquisition, field investigation, formal analysis and visualization. BV participated to the field investigation, formal analysis and visualization. EW and WA participated to the funding acquisition and supervision. All authors contributed to the manuscript preparation.

## 11. Competing interests

The authors declare that they have no conflict of interest.

## 12. Appendix

**Table A1: Compaction_Daily table.** Stores daily compaction records for each FirnCover instrument.

| Field Name | Comments |
| --- | --- |
| sitename | Name of the FirnCover site |
| daynumber_YYYYMMDD | Year, month, date of the measurement |
| Compaction_Instrument_ID | linked to "Compaction_Instrument_Metadata" table |
| Compaction_Ratio_Med | The ratio of the compaction line measurement (fraction of total instrument cable length), values 0 to 1, inclusive. Uses a median value of six daily measurements. |

| Compaction_Wire_Correction_Ratio_Med | The ratio of the wire resistance as a fraction of the total line resistance. Values 0 to 1, inclusive (typically below 0.001). |
|---|---|
| Compaction_Cable_Distance_m | Distance the instrument wire is extended, typically 0-2 m (up to 5 m for extended-cable instruments) |
| Compaction_Borehole_Length_m | Length of the borehole at that time step. Combines the updated cable length with the initial borehole length. |
| Borehole_Depth_Top_m | Depth from the surface to the top of the borehole at that time step, combining the initial borehole depth (0 for the surface) and the Sonic Ranger snow depth measurement. |
| Borehole_Depth_Bottom_m | Depth from the surface to the bottom of the borehole. Computed as "Borehole_Depth_Top_m" + "Compaction_Borehole_Length_m" |

**Table A2: Air_Temp_Hourly table**

| Field Name | Comments |
|---|---|
| sitename | Name of the FirnCover site |
| daynumber_YYYYMMDD | Year, month, date of the measurement |
| hournumber_HH | Hour of the day (0 through 23) |
| AirTemp_C | ~2 m air temperature at that hour, measured by the shielded L109 thermistor, in °C. Actual height of the temperature measurement can be derived by adding 28 cm to the Sonic Ranger height in "FirnCover_Meteorological_Daily_DataTable" |

**Table A3: Meteorological_Daily table**

| Field Name | Comments |
|---|---|
| sitename | Name of the FirnCover site |
| daynumber_YYYYMMDD | Year, month, date of the measurement |
| BattV_min_V | Minimum station battery voltage for the day |
| BattV_max_V | Maximum station battery voltage for the day |
| PanelTemp_mean_C | Mean daily temperature (°C) measured on the data logger inside the logger box |
| AirTemp_min_C | Minimum daily air temperature (°C) measured hourly |
| AirTemp_max_C | Maximum daily air temperature (°C) measured hourly |
| SonicRangeQuality | The quality score value of the Sonic Ranging sensor, chosen as the highest-quality of 24 daily |

| | measurements. Ranges from 162 to 600 with good quality scores below 210. |
|---|---|
| SonicRangeQualityCode | 0=Good, 1=Questionable, 2=Poor, 3=No Measurement |
| SonicRangeDist_Raw_m | The raw distance measured by the sonic ranger, before temperature correction. |
| SonicRangeAirTemp_C | The air temperature at the time of the sonic ranger measurement. |
| SonicRangeDist_Corrected_m | The corrected distance measured by the sonic ranger. |
| Accum_Snow_Depth_m | The accumulated snow depth since the instruments' installation, corrected for tower raises upon revisits. |

**Table A4: Firn_Temp_Daily table**

| Field Name | Comments |
|---|---|
| sitename | Name of the FirnCover site |
| daynumber_YYYYMMDD | The day of the reading |
| RTD_Ohms_Avg | Average RTD resistance reading |
| RTD_Ohms_Max | Maximum RTD resistance reading |
| RTD_Temp_Avg_Uncorrected_C | Average RTD temperature reading (deg C) |
| RTD_Temp_Max_Uncorrected_C | Maximum RTD temperature reading (deg C) |
| RTD_Line_Correction_Ohms_Avg | The line correction |
| RTD_Temp_Avg_Corrected_C | Average RTD temperature reading (deg C), w/ adjustment for wire resistance |
| RTD_Temp_Max_Corrected_C | Maximum RTD temperature reading (deg C), w/ adjustment for wire resistance |

**Table A5: Station_Metadata table**

| Field Name | Comments |
|---|---|
| sitename | Name of the FirnCover site |
| iridium_URL | The online URL where transmissions are collected |
| latitude | The WGS84 latitude of the station upon installation |
| longitude | The WGS84 longitude of the station upon installation |
| installation_daynumer_YYYYMMDD | The day the station was installed. |
| comments | General comments about the station upon its installation. |
| RTD_stringnumber | The string serial number of the RTD string installed at the station. |
| RTD_installation_daynumber_YYYYMMDD | The day at the RTD was installed at the station. |
| RTD_top_usable_RTD_num | Number (from the top) of the first usable RTD sensor. Non-usable sensors could not be inserted in the snow and were left lying on the surface. |

| | |
|---|---|
| RTD_depths_at_installation_m | The 24-length depths of each RTD at installation. |
| RTD_direction_from_tower_degrees | The compass direction (non-corrected for declination) of the station tower to the RTD string. |
| RTD_distance_from_tower_m | The distance from the station tower to the RTD string. |

**Table A6: Station_Visit_Notes table.**

| Field Name | Comments |
|---|---|
| sitename | Name of the FirnCover site |
| daynumber_YYYYMMDD | Day of the visit |
| visit_notes | Notes about the site visit or revisit. |

**Table A7: Compaction_Instrument_Metadata table.** Installation depths and positions of each FirnCover compaction instrument.

| Field Name | Comments |
|---|---|
| instrument_ID | Unique identification number of the instrument |
| sitename | Name of the FirnCover site |
| installation_daynumber_YYYYMMDD | Date that the instrument was installed. |
| borehole_top_from_surface_m | The top of the borehole from the surface upon installation, in m. (0 for surface, negative numbers for beneath the surface) |
| borehole_bottom_from_surface_m | The depth from the surface to the bottom of the borehole, in m |
| borehole_initial_length_m | The length of the borehole, in m |
| instrument_has_wire_correction | Whether the instrument installed has a wire-resistance correction sensor installed, or not. |
| direction_from_tower_degrees | The compass direction (not corrected for declination) from the tower to the instrument. |
| distance_from_tower_m | The distance (in m) from the tower to the instrument. |
| borehole_ID | The identifying name of the core taken from the borehole, where stratigraphy and density were measured (names consistent with cores in the NASA SUMup dataset). |
| borehole_ID_is_direct | A "direct" (True) core density profile came straight from that borehole. If "indirect" (False), that core was not profiled for density directly, and this links to a nearby. Adjacent core measured at the same time, typically within 10-20 meters distance. |

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
