# Peer review of "The Greenland Firn Compaction Verification and Reconnaissance (FirnCover) Dataset, 2013-2019"

_Earth System Science Data, 2021_

## Author Comment (AC1)

Dear reviewers and editorial team,

We thank the three reviewers for very constructive comments that improved the manuscript.

We addressed these comments and suggestions below. The reviewer's comments are in black while our responses are in green and quotes from the revised manuscript are in *italic green*.

Sincerely,

Baptiste Vandecrux on behalf of the authors.
* * *
**Ian McDowell:**

MacFerrin et al. describe a dataset consisting of strain rates in shallow firn, 2 m air temperature, firn temperature, and surface height from eight sites across the Greenland Ice Sheet. The dataset consists of daily strain measurements and hourly meteorological measurements spanning 2013 – 2019. Multiple boreholes have been instrumented at each site allowing for users to assess viability and repeatability of strain measurements and examine compaction rates over different depth ranges.

Given the disagreement in firn models using both steady-state and transient modes, this dataset will provide important validation data spanning climate regimes. The dataset will provide opportunity for interesting and important future work examining firn compaction on seasonal timescales and investigating the effects of meltwater on firn compaction rates.

This paper is well-written and provides a concise but helpful literature review in addition to describing the dataset. I have a few general comments and questions along with some technical corrections that are detailed below, but I recommend this paper be published in ESSD after these are addressed.

**General Comments:**

**(1)**

The authors provide a nice summary of the recent research on Greenland's firn layer. I have one suggestion that may make their discussion clearer. The authors state four main reasons for studying the firn layer: (1) to refine estimates of mass balance using altimetry methods; (2) to estimate how much meltwater can be stored within the firn column and buffer future sea level rise; (3) to understand the development of near-surface ice slabs that block future meltwater percolation; and (4) to refine interpretations of climate records from ice cores.

In my opinion, numbers 2 and 3 essentially fall into the same category of understanding firn's capacity to buffer future sea level rise. I would recommend phrasing this section as there being 3 main reasons for studying firn: (1) to use altimetry mass balance products; (2) to understand how much water can be stored in the firn column and buffer sea level rise – can all available pore space be filled or does the expansion of near-surface ice slabs block percolation and expand runoff zones? And (3) to

improve interpretations of climate records contained in ice cores. Grouping the main reasons for studying firn in this way seems to make more intuitive sense to me and better links the impacts of ice slab formation on meltwater storage.

Additionally, I would consider adding an additional phrase or sentence to expand on why firn structure is important for interpreting paleoclimate records. I would recommend just explicitly stating that knowing the ice-age/gas-age discrepancy allows us to accurately date past atmospheric conditions. This last reason just seemed a bit shorter than the others listed.

Thank you for your suggestion. We rephrased to:

*The GrIS's firn layer has been the subject of recent research for multiple reasons. First, [...]. Second, the firn is able to retain part of the meltwater generated at the surface and buffer sea level rise (Pfeffer et al., 1991). The firn's retention capacity depends on: i) the pore volume available for meltwater storage (Harper et al., 2012), which is decreasing (Vandecrux et al, 2019); ii) the firn's cold content, which is the energy required to bring the firn to the melting temperature (Vandecrux et al., 2020); and iii) on the capacity for the meltwater to reach depths where retention is possible, which is for example reduced in presence of low-permeability near-surface ice slabs (Machguth et al., 2016, MacFerrin et al. 2019). Third, the firn impacts climate records preserved in ice cores. Bubbles of atmospheric gasses become trapped in closed pores at the firn-ice transition, and knowledge of the age of the firn at this bubble close-off depth is essential to accurately establish the chronology of past climate changes (Schwander and Stauffer, 1984; Schwander et al., 1997).*

**(2)**

I am curious about the contribution of ice flow to vertical velocities that could be captured by the installed strainmeters and misinterpreted as firn compaction. If the ice column were undergoing longitudinal extension, the borehole could be shortening even in absence of firn compaction. I imagine that at sites like Summit, there likely is not much longitudinal compression, but with EastGRIP being located on an ice stream and there being local topographic variability at sites like Crawford Point, I am wondering how much ice dynamics could affect the calculated firn compaction rates. In studies that attempt to determine vertical strain at sites using phase-sensitive radar, some have attempted to separate out an ice dynamics component from compaction (e.g. Jenkins et al., 2006).

My two main questions arising from this curiosity are: (1) Would vertical strain resulting from ice flow be recorded by your strainmeter instrumentation? and (2) If so, are there existing data that users of your dataset could find to remove an estimated vertical strain component from ice dynamics to get a purely firn compaction component?

If the answer to the first question is *no*, then perhaps adding sentence or two explaining the concern and detailing that this instrumentation would not be affected by ice dynamics processes would help readers. If the answer is *yes*, then I think it would greatly help users of your dataset to point to existing

measurements or potential models that could help resolve the influence of ice flow on the compaction rates that you present.

Thank you for this comment that opens an unexplored facet of firn dynamics. We added the following paragraph to the discussion:

*It is possible that our compaction measurements could be affected by horizontal divergence (Horlings et al., 2021). However, for the present analyses, we consider these effects to be negligible, which is consistent with firn-densification modeling efforts in Greenland (Kuipers-Munneke et al., 2015). A more thorough analysis could use ice velocity measurements (e.g. Joughin et al., 2016) to explicitly account for the effects of ice flow.*

**(3)**

I am interested in knowing the distance between boreholes at each site. It may be helpful to include a sentence on line 111 saying that boreholes are spaced approximately xx m apart. If available, could each borehole's coordinates be included in Table 2? It may make Figure 1 too busy, but potentially including insets for each site with the locations of each borehole may also be helpful.

Thank you for your suggestion. However we find several drawbacks in the presentation of all the instruments location. Table 2 does not have more space for additional columns. We also think that a list of coordinates does not tell much to the reader. They would then need to copy that table in their GIS software to inspect the instruments' position. We also think that plotting it in Figure 1 would be too much information. However, we do indicate in Table 3 that there is a data table called "Compaction_Instrument_Metadata" where the user will find the instrument's direction (in compass degrees) and distance (in m) from its tower.

We also added the following general statement at line 111:
The instruments were generally within 10 m of the tower and their position relative to the tower are given in the table Compaction_Instrument_Metadata (Table A7).

This interests me because looking closely at Figure 5, it appears there are some slight differences in compaction rates between boreholes at the same site that are installed over approximately the same depth range. Specifically, I am looking at boreholes 13 and 16 at NASA – SE in the summer of 2015, boreholes 17 and 20 at Saddle, boreholes 30 and 33 at Summit (particularly in the summer of 2015), and boreholes 26 and 29 at EastGRIP. For future work, it would be interesting to examine discrepancies in compaction rates between closely-spaced boreholes instrumented over the same depth range, and knowing the length scale over which these differences occur would be useful.

Indeed this is an interesting question. However, the study of the spatial scaling of firn densification will also require the analysis of spatial distribution of firn ice content, density and temperatures. This is a complex analysis that is beyond the scope of the present data-focused paper.

**(4)**

Suspicious data readings that are not included in the preliminary analysis are mentioned on lines 210 – 211. Could a brief explanation as to why they are considered suspicious be provided here?

We added the following:

*For a number of the instruments, there are periods of data that we consider suspicious because of abrupt jumps in the compaction rates. We hypothesize that this could be due to ice buildup on the cable that prevented the instrument from working, and once the cable became free the instrument began to work again.*

**Minor Comments and Technical Corrections:**

**L18:** Consider changing *"snow temperature"* to *"firn temperature"* or *"snow and firn temperature"*

Corrected, thank you.

**L42:** "*Adolf and Albert, 2014"* should be changed to "*Adolph and Albert, 2014"*

We removed this reference and substituted a paper that is more focused on delta age from a firn-densification model.

**L82:** Does there need to be a dash here in *"firn-density" ?*

We are not sure where the reviewer is referring to - there is no mention of 'firn-density' on line 82. We have tried to use hyphens consistently in the cases where there is a compound adjective, e.g. "firn-density profile" should have a hyphen.

Throughout the manuscript, *"SumUp"* is written when I believe the correct capitalization is *"SUMup"* .

Corrected, thank you.

Throughout the manuscript text and figures, EastGRIP is written both as *"EastGrip"* and *"EastGRIP"*. I recommend that it be written as EastGRIP throughout the manuscript.

Corrected, thank you.

**L256:** Does there need to be a dash here in *"firn-temperature"* ?

Yes; however, we changed the sentence to: *The FirnCover dataset also includes measurements of air temperature, surface height, and, at all sites except EastGRIP and NASA-SE, firn temperature*

**Figure 3:** The caption mentions *"changes relative to installation depth plus 120 days"*, however, in line 195 you suggest removing the first month of data from analysis. In this figure, have approximately the first 4 months of data been removed? It may be helpful to state this in the text, perhaps just after mentioning that the first month of data has been removed but Figure 3 shows data with the first four months removed.

Thank you for spotting this. This was actually two mistakes, since we discarded the first 60 days of measurements. We rephrased, in the methods section, how we deal with the settling period:

*At most strainmeters, the first weeks to months of record show relatively high compaction rates. This initial period of increased compaction is more pronounced for instruments installed at the surface*

*than the ones installed at the bottom of a snow pit (Table 2). We consider these high initial compaction rates to be the result of the instrument settling over the snow and firn. This period of initial settling needs to be discarded to study the firn after it adapted to the presence of the instrument. At KAN_U, where the deeper firn is rich in ice (Figure 3), settling of the instrument is mainly due to the surface snow and took about a month. At Summit, where the firn has no ice layers, settling took about two months. For this preliminary analysis, we discard the first 60 days of recordings for each instrument, but a site-specific analysis of instrument settling may allow the recovery of more observations within that period.*

We also removed the mention of the settling period from the caption of the Figure 4 (previously 3), since the figure only shows the valid data (without the settling period).

**Figure 6:** It appears that the firn temperatures at Summit are not captured by the scale bar on the right. I think the scale bar needs to be expanded to capture the low temperatures of firn at Summit. Additionally, the color scale used in this figure is slightly counterintuitive, as I often think of red colors as warmer temperatures. Perhaps flipping the scale may help readers interpret it better.

Thank you for the suggestion. We updated both the scale and the color map.

**Table 4:** I think the temperature difference column should be rounded to the same number of decimal places as the previous two columns. If there is not enough certainty to present measurements with two or more decimal places, it seems odd that the difference between the two is presented at a higher resolution than the measurements.

Agreed. This was corrected.

**Reference included in review comment:**

Jenkins, A., Corr, H. F., Nicholls, K. W., Stewart, C. L., & Doake, C. S. (2006). Interactions between ice and ocean observed with phase-sensitive radar near an Antarctic ice-shelf grounding line. *Journal of Glaciology, 52*(178), 325-346.
* * *
**Joel Harper:**

The dataset presented in this paper is novel and resulted from a large work effort. These data are therefore important and have high value to the scientific community. The manuscript is well-written, well-organized, and is comprehensive. I support publication of this paper in ESSD as a strong contribution to the literature. I have a small number of minor comments.

Comments

line 85: I believe the 'is' should be 'are' to match plurality.

It was unclear what "is" referred to. We reformulated to:
*Finally, some densification models are tuned to match firn-density observations while forced by RCM-simulated surface forcing. The biases that may exist in that surface forcing are then*

*compensated by the tuning of the densification model, which can then give inappropriate response under a different climate forcing.*

Line 134: I was confused about numbers here. Understand that it was a lot of work for a failed installation, but it seems much more straightforward to just tell us about the 48 sensors with actual records.

Agreed. We now only mention 48 instruments.

Line 140: could report manufacture's specs on strain of the wire. Over these short runs I would assume it's a non-issue?

Thank you for this question that triggered a revision of our expected accuracy.

The potentiometer has an accuracy of ± 0.1 % (±2 mm when fully extended). Typical elongation for vectran string under these loads is less than 0.15 % (15 mm for a 10 m extension). So we now summarize the instrument accuracy as:

*Including potentiometer accuracy and minimal elongation of the extended string, measurement uncertainty on the borehole length is within ± 2 cm. Measurement of the borehole shortening, however, is insensitive to the elongation of the wire that is under a constant load and can be made with an accuracy of ±2 mm.*

Line 145: if you excavated the PVC platforms, it would be nice to know how they settled with time: i.e., did they tilt heavily to one side?

We would like to know, but we did not dig out to inspect how the platform settled. The settling is visible in the data and we give recommendation on how to deal with it:

*At most strainmeters, the first weeks to months of record show relatively high compaction rates. This initial period of increased compaction is more pronounced for instruments installed at the surface than the ones installed at the bottom of a snow pit (Table 2). We consider that these high initial compaction rates are the result of the instrument settling over the snow and firn. This period of initial settling needs to be discarded to study the firn after it adapted to the presence of the instrument. At KAN_U, site where the deeper firn is rich in ice (Figure 3), settling of the instrument is mainly due to the surface snow and took about a month. At Summit, where the firn has no ice layers, settling took about two months. For this preliminary analysis, we discard the first 60 days of recordings for each instrument, but a site-specific analysis of instrument settling may allow the recovery of more observations within that period.*

Line 157: the approximate depth, or range of depths, of the snowpit installations would be good information to provide.

*This information was already included in Table 2. We added:*

*(see non-zero initial depth of borehole top in Table 2)*

Section 3.2

-while the air temp appears to have been measured hourly, the firn temps were measured once per day? Clarification would be useful because some users may be interested in high time resolution firn temps.

*Agreed. We added:*

*Unlike air temperature, surface height and firn temperature are available as daily averages.*

-were the firn temperature holes backfilled (if so, how?) or left standing air filled? This is important information to include for future data users.

*Agreed. We added:*

*The RTD strings were installed in separate boreholes that were backfilled with snow.*

Line 170: 'clearing chips' will confuse some readers. "due to accumulated drill shavings on the bottom of the hole" …or some such.

*Thanks, we added:*

*due to accumulated drill shavings at the bottom of the boreholes.*

Line 195-paragraph: appreciate this approach/section.

Figure 3:

-"…depth plus 120 days" is awkward wording that hung me up (L+T ?).

*Indeed that was confusing. We removed this mention of the discarded settling period in the caption.*

-The caption needs clarification of what the legend numbers/lines are …e.g., "legend is sensor number in Table X".

*Thanks, updated. We also added the initial length of the borehole as requested by another reviewer.*

Line 244: recognize that the scope of this paper is limited, but you mention a hypothesis that noise stems from interaction between meltwater and the borehole. This could use a bit more elaboration – meltwater dribbling down the hole somehow generates a noisy signal?

*We realized that we do not have strong evidence about the origin of this noise (or even if it is actually noise). We removed this rather hypothetical paragraph.*

Figure 4: as w/Fig 3, the caption could use explanation of the legend numbers.

Thanks, updated.

-Overall, an enjoyable read and high value contribution.

Thank you!
* * *
**Megan Thompson-Munson:**

This manuscript introduces FirnCover, which is a unique dataset of firn compaction measurements taken between 2013 and 2019 at eight locations on the Greenland Ice Sheet. MacFerrin et al. detail the methods used to obtain the measurements, describe the format of the dataset, and include a preliminary analysis and interpretation of the data. Data from strainmeters, air temperature sensors, and firn temperature sensors at eight sites with different climates provide the opportunity to investigate variability in firn compaction based on climatic factors. Moreover, this dataset of in-situ observations from 50 strainmeters can act as a comparison for existing firn models, which the authors point out use different densification schemes and often disagree with one another. FirnCover is a novel, high-quality, and well-prepared dataset that will be a valuable contribution to the firn modeling and observation community.

Overall, this paper is very well-written and the dataset is of high quality. Included are two general comments as well as line-by-line technical corrections. Once these comments have been addressed, I recommend this manuscript be published in Earth System Science Data.

**General Comments**

**1) Data overview and preliminary analysis** - The data description and analysis (Section 5) is important to include in the manuscript since it describes and shows the actual data included in FirnCover. Since the goal of the paper is to present the dataset, finding a balance in the amount of preliminary analysis to include is difficult, and I think that toning back some of the analysis (or even including more of the raw data) could help reach this balance. This section could benefit from showing more of the raw data and focusing less on the analytical choices (e.g., averaging window, amount of smoothing, how much initial data to ignore), especially since the authors have included the full record of unsmoothed data in the publicly available dataset (a very good decision). For example, in Section 4, lines 192-198, the authors describe how the dataset includes the first month of measurements, but those data have been discarded for the analysis presented in the manuscript. Since users of the dataset will likely have to decide on how to filter these data for their own analysis, showing a figure of the full, unsmoothed record could be beneficial. This could even just be for one example record rather than all of them.

Thank you for this suggestion. Indeed the choice of what goes into the analysis is an open question. By focusing on a higher-level analysis and by not displaying the raw data, we want to show the potential of our dataset after a very simple and accessible post-processing (analysis and plotting scripts are also made available). We do not think that showing the raw data will help to build trust and user community. Some modelers that want to evaluate their densification model will look directly for time series such as shown in Figure 3 and 4 and will want ancillary data like displayed in Figure 4 and 6. We do not think that a detailed analysis of the first 60 days of not-so-good measurements will be interesting and rather focus on the rest of the dataset.

**2) Interpretation** – On a similar note, I would consider adjusting the amount of interpretation included in Section 5. There are a few instances where the authors state the reasons for higher firn density without referencing a figure or table (e.g., line 223-224), and it's unclear whether these interpretations are based on the data itself or knowledge from prior studies. If based on the data, specifically referencing figures that support the interpretation will make the claim stronger. More specific examples regarding this comment have been listed in the line-by-line section below.

Thank you for pointing this out. Indeed the density information is closely related to the FirnCover data although it is located in a different data repository (SUMup). We now illustrate the density profiles at each site in Figure 3.

**Minor Comments and Technical Corrections**

Lines 53-58: It seems that this section might need a citation to better support these statements.

Indeed this part lacked references to previous work:

*Meltwater refreezing increases the firn density when surface meltwater or rain refreezes in the firn's pore space (e.g. Braithwaite et al., 1994; Reeh, 2008). This occurs primarily in the warmest regions of the ice sheet's accumulation area. The two above-mentioned phenomena are interconnected because meltwater refreezing releases latent heat and increases the firn temperature, which accelerates compaction of surrounding firn (Pfeffer and Humphrey, 1996; Humphrey et al., 2012). In the highest-elevation zones of the ice sheet, where firn densification mainly occurs through compaction, the compaction rate in the near-surface firn varies seasonally due to the fluctuating temperature; the deeper firn does not experience this seasonal variation in compaction rate (e.g. Herron and Langway, 1980; Arthern et al., 2010; Ligtenberg et al., 2011; Morris and Wingham, 2014).*

96: Consider expanding the description of these existing measurements and maybe even include them in Figure 1. If they are in regions or climates that are different from those in FirnCover, this would help amplify the need for FirnCover.

Thank you for this suggestion. We gave expanded this description:

*The uncertainties associated with firn-model development and the disagreement among the existing models underscore the need for direct measurements of firn compaction. The direct observation of firn compaction implies either tracking the thickness of a portion of firn (Hamilton et al., 1998; Arthern et al., 2010), the optical tracking of layers in a borehole (Hubbard et al., 2020) or the tracking of layers in repeated high-resolution density profiles (Morris and Wingham, 2014). Most of the firn compaction measurements have been conducted in Antarctica (Hamilton et al., 1998; Arthern et al., 2010; Hubbard et al. 2020). Lastly, the only firn compaction measurements available in Greenland (Morris and Wingham, 2014) derived average compaction rates over specific periods spanning from 2004 to 2011 and over a single transect in central western Greenland.*

140: To what is the 2.032 m range referring? The lengths of the boreholes are several meters, so is this range the amount of length change that can be detected?

This number is the winding capacity of the potentiometer. If the boreholes shorten more than that distance, the reel is full, and the potentiometer cannot measure anymore. Since none of the boreholes shortened by more than ~1.2 m (Figure 3), this information is not important for the data user. It can anyway be found in the technical documents of the potentiometer, which model and manufacturer are given in the text.

192: Wouldn't the instrument settling depend on the snow surface density, such that the lower the snow density, the longer the settling time? If I'm interpreting this correctly, wouldn't the one-month settling time of the high-density ice site be the minimum settling time among all sites? This seems to be mentioned later in lines 239-241. Consider combining this discussion or changing it to discuss how surface density affects settling rates, and the settling period is likely different for each site.

We now rephrased our approach to the settling period and why we now use 60 days:

*At most strainmeters, the first weeks to months of record show relatively high compaction rates. This initial period of increased compaction is more pronounced for instruments installed at the surface than the ones installed at the bottom of a snow pit (Table 2). We consider these high initial compaction rates to be the result of the instrument settling over the snow and firn. This period of initial settling needs to be discarded to study the firn after it adapted to the presence of the instrument. At KAN_U, where the deeper firn is rich in ice (Figure 3), settling of the instrument is mainly due to the surface snow and took about a month. At Summit, where the firn has no ice layers, settling took about two months. For this preliminary analysis, we discard the first 60 days of recordings for each instrument, but a site-specific analysis of instrument settling may allow the recovery of more observations within that period.*

We also note that the Figure 4 in the first version of the manuscript did not have the settling periods discarded. In the revised manuscript, that figure (now Figure 5) do not show these initial high compaction values, and allows better visualization of the seasonal (and hopefully natural) variation of the compaction rates.

204: Explain why a period of two months is chosen for the running mean. Users of the dataset may be looking for guidance in processing/analysis, so a description of the authors' reasoning may help.

Unfortunately, the amount of filtering needed is still open. Two months may be too much for certain sites (removal of weekly natural variations) and too little at others (remaining instrumental noise). That is why we encourage the data users to develop their own filtering strategies in the following sentences. We changed the final sentence in that paragraph to: *The dataset includes the unfiltered data, and we recommend that users apply their own filtering strategy specific to their needs.*

211: Why are these data suspicious? Consider elaborating on the suspicious nature or showing them in a figure

We elaborated as follows:

*For a number of the instruments, there are periods of data that we consider suspicious because of abrupt jumps in the compaction rates. We hypothesize that this could be due to ice buildup on the cable that prevented the instrument from working, and once the cable became free the instrument began to work again.*

223: Change "this is due" to "this is likely due" and consider adding a citation or referencing a figure to back this claim.

This sentence was removed.

Figure 3: The first line of the caption is confusing. The current wording makes it seem as though time is being added to depth. Is this record starting 120 days after the installation *date*? If so, why?

Indeed that was confusing. We removed this mention of the discarded settling period in the caption.

240: Consider changing "deformation" to "settling"

Thanks, updated.

244: Is this really "noise" or this is a result of the densification processes mentioned? This is an area where I would be cautious of over-analyzing/interpreting for the purposes of this paper.

We agree that we do not have strong evidence about the origin of this noise (or even if it is actually noise). We removed this rather hypothetical paragraph.

246: Earlier it was stated that KAN-U is completely ice (line 194), but here the authors write that it contains a 5-m ice slab. Consider updating either line 194 or 246 to reflect the same meaning.

This point should now be clarified with the density profiles being plotted in Figure 3. The line 194 was also rephrased.

249: Reference Figure 4 at the end of the sentence.

Updated.

249-251: The wording in the sentence is confusing. Remove the "(resp. __)" and split into two distinct sentences.

Updated.

Figure 4: Since there is dependency on borehole depth (line 237), could you also label the different lines by depth (in addition to instrument ID)? This could also apply to Figure 3.

Figure 3 and 4 were updated after your suggestion.

257: Consider changing "compaction rates" to "surface height" since the referenced figures are showing surface height.

We updated this sentence to:
*...these data enable us to relate the compaction rates (Figure 4 and 5) to the surface and subsurface conditions (Figures  6 and 7).*

Figure 6: It seems these firn temperatures are interpolated; that should be mentioned in the caption. Also, it seems there was a different color bar used in the Summit plot. Consider flipping the color bar so warmer=redder, or consider using a perceptually uniform sequential color map (e.g., viridis) that has a larger range of colors and can visually capture small variations better.

We updated the caption, colormap and colorbar range. Thanks for your suggestion.

268-270: Provide a clearer description of what two variables are being compared. Over what period is the average air temperature calculated? Is the firn temperature averaged over 10 m and over time, or is it the firn temperature taken at 10 m and averaged over time?

We added details about these two variables:

*For each site, we interpolate the 10 m firn temperature and average it over the 2015-2019 period covered by the measurements (Table 4). We compare this firn temperature to the average air temperature calculated for the years where more than 90% of the temperature measurements are available (Table 4). The average air temperature and interpolated 10m firn temperatures are rarely equivalent (Table 4).*

268-270: I wonder if this comparison would have more significance if examined seasonally rather than averaged over time. This is an area where I would consider being cautious with the interpretation since it seems to require a more in-depth analysis, which is not necessarily the focus of this manuscript.

Indeed, there is a lot more to investigate from this data. The idea of this preliminary analysis is to present the data in a simple way so that it triggers interest and usage by the community. We hope that the data are used for seasonal analyses in future research.

284-287: Consider adding this information to either Table 1 or 2 so the reader can easily see what existing data are available at each FirnCover site.

Thank you for your suggestion. In the summary remarks, we wanted to remind the reader that other datasets exist at our study sites. We do not think that our manuscript (that we want focused on our data) should give an exhaustive list of other measurements conducted at our sites, which include well-studied locations such as Summit, EastGRIP or Dye-2, where the list would be very long.

---

## Author Response (AR2)

Response to topical Editor Ge Peng

> Line 140: "A total of 48 strainmeters were installed at the 8 FirnCover stations."
> Please double-check the accuracy of this statement, if I understand correctly, a total of 50 strainmeters were installed at the 8 FirnCover stations but you are only using data from 48 because the other two did not take any measurements for the entire period.
> Given the comment by Dr. Harper, perhaps you can state that your dataset is based on the measurements from 48 strainmeters installed at the 8 FirnCover stations.

Thank you for the thorough look at our revised manuscript. We changed the sentence l.140 to:
We here present data from 48 strainmeters installed at 8 FirnCover stations.

Baptiste Vandecrux on behalf of the authors